# TASK-RELEVANT FEATURES OUTPERFORM LEARNED REPRESENTATIONS FOR DRUG-MICROBIOME RETRIEVAL

## 1 INTRODUCTION

Biological foundation models have shown mixed results: large-scale pretraining enables strong generalization in some domains while underperforming simpler baselines in others (Wu et al., 2025; Feng et al., 2025). Understanding when and why these gaps arise remains an open challenge. We present a case study from drug–microbiome interaction, where the gap can be traced to a specific cause: the choice of input features.

Non-antibiotic drugs act on microbial enzymes and metabolic pathways rather than taxa directly, yet drug–microbiome studies typically use genus-level abundances as the primary representation (Nguyen et al., 2023). PICRUSt2 (Douglas et al., 2020) offers an alternative: it infers enzyme commission (EC) profiles from 16S amplicon data via a deterministic, many-to-many mapping from genera to enzymes. Because multiple genera encode the same enzyme and each genus encodes many enzymes, EC profiles may group drugs that perturb different taxa but the same pathways. We test whether this biologically-informed feature choice matters more than the compression method (PCA vs VAE) or pretraining scale (MGM, 263K samples). It does.

## 2 METHODS

**Data.** 684 drugs were screened on SIC0, a synthetic intestinal community (Shi et al., 2025), with 2,231 treated and 149 control samples profiled by 16S amplicon sequencing. SIC1–8 comprised eight additional communities (88 drugs each). We computed delta profiles by applying CLR transform to abundances (101 genera; 2,538 ECs from PICRUSt2) and subtracting control centroids. See Appendix A for full data pipeline. **Models.** We compare PCA (10-d), matched $\beta$-VAEs (10-d latent), and MGM (Zhang et al., 2025), a transformer pretrained on 263K microbiome samples. See Appendix B for architectures. **Evaluation.** Each representation embeds samples into a vector space; we average replicates to obtain one embedding per drug. For each query drug, we retrieve the 10 nearest neighbors and compute Mean Average Precision (MAP@10) based on Anatomical Therapeutic Chemical (ATC) class agreement at levels 1 and 2. We evaluate within-community (70/30 split on SIC0) and cross-community (leave-one-community-out CV on SIC1–8). See Appendix C for details.

## 3 RESULTS

Within-community rankings differ from cross-community rankings, but LOOCV reveals a stable pattern (Table 1).
Within SIC0, all confidence intervals overlap—no representation stands out. Across communities, a single clear pattern emerges: the six taxonomy- and EC-based representations are indistinguishable (all pairwise $p > 0.35$ at ATC level 2), while MGM falls clearly below ($p \leq 0.02$). Reducing MGM to 10-d via PCA does not close the gap, ruling out a dimensionality explanation.

**Why are EC profiles more conserved?** Across communities, EC delta profiles are significantly more similar than taxonomy profiles (mean cosine distance 0.72 vs 0.84; Wilcoxon $p = 3.5 \times 10^{-12}$, Cohen's $d = 1.09$). This follows from PICRUSt2's many-to-many mapping: when communities swap genera that encode the same enzymes, EC profiles are preserved but taxonomy profiles change. Figure 1 shows the pattern at the drug level: most drugs (25/34) have more conserved EC profiles,

| | Within-community | | LOOCV (8 folds) | |
|---|---|---|---|---|
| Representation | ATC L1 | ATC L2 | ATC L1 | ATC L2 |
| EC PCA (10-d) | .223 | .113 | .409 ± .024 | .479 ± .059 |
| EC VAE (10-d) | .225 | .097 | .412 ± .033 | .476 ± .045 |
| Raw EC | .222 | .088 | **.429** ± .021 | **.506** ± .047 |
| Taxa PCA (10-d) | .278 | .102 | .407 ± .049 | .496 ± .045 |
| Taxa VAE (10-d) | .297 | **.115** | .410 ± .038 | .476 ± .057 |
| Raw Taxonomy | .296 | .090 | .394 ± .035 | .482 ± .041 |
| MGM (256-d) | **.302** | .075 | .374 ± .040 | .427 ± .050 |
| MGM→PCA (10-d) | .282 | .117 | .382 ± .044 | .405 ± .061 |

Table 1: MAP@10 across two evaluation regimes. Within-community: 70/30 held-out split on SIC0 (all CIs overlap). LOOCV: leave-one-community-out on SIC1–8 (mean ± std over 8 folds). Among the six non-MGM representations, all pairwise differences are non-significant ($p > .35$ at ATC L2). MGM trails significantly: EC PCA vs MGM $p = .019$, vs MGM→PCA $p = .002$ (paired bootstrap over folds).

but the 9 where taxonomy is more conserved are enriched for antibiotics—where killing specific taxa *is* the signal. Notably, nitroimidazoles (metronidazole, tinidazole, ornidazole) cluster with EC-conserved drugs despite being antibiotics; these are prodrugs requiring metabolic activation, so their mechanism is functional rather than taxonomic. This selective failure rules out a pure compression artifact: if EC's tighter clustering were merely collapsing information, it would affect all drugs equally. Per-community MAP@10 confirms this: EC performance correlates less with genus overlap ($\bar{\rho} = 0.13$) than taxonomy performance does ($\bar{\rho} = 0.40$).

## 4 DISCUSSION

The central finding is that **input features matter more than model architecture or scale**. Raw features, PCA, and VAEs are interchangeable in LOOCV—but the pretrained foundation model trails all of them. Neither nonlinear compression nor large-scale pretraining adds value when task-relevant features are already available.

**Context-dependence.** MGM leads within-community at ATC level 1 but trails cross-community. Taxonomy VAE leads within-community at ATC level 2 but shows no advantage in LOOCV. The ranking changes because the task changes: within-community evaluation rewards any signal, while cross-community evaluation specifically rewards features that transfer. Domain knowledge encoded in the feature pipeline (PICRUSt2 + CLR + delta) transfers; learned representations from a general-purpose model do not.

**Limitations.** SIC0 was reserved for hyperparameter selection; LOOCV uses only SIC1–8 to avoid leakage from architecture tuning. LOOCV uses seen drugs (community generalization, not drug generalization). Test sets are small (∼35 drugs at level 2 per fold). Synthetic communities with near-complete PICRUSt2 coverage represent a best case. ATC classes are a coarse proxy for mechanism.

**Future work.** PICRUSt2's taxa-to-enzyme mapping relies on reference genomes, limiting coverage for novel taxa. Tools like MarkerMAG (Song & Thomas, 2022) can link 16S ASVs to metagenome-assembled genomes (MAGs), enabling direct functional annotation of contigs via gene prediction. Systematically applying this to public metagenome datasets could yield improved mappings that preserve PICRUSt2's many-to-many structure while extending coverage and accuracy.

MEANINGFULNESS STATEMENT

What makes a representation of life meaningful? We argue the answer depends on the question being asked. The same microbial community can be described by its genera, its enzymes, or its learned embedding—all faithful, none uniquely correct. We show that genus-level and enzyme-level features both outperform a foundation model trained on 1,000× more data for drug-mechanism retrieval across communities. This demonstrates that meaningful biological representations are not defined by model scale or expressiveness alone, but by alignment with the invariances required by

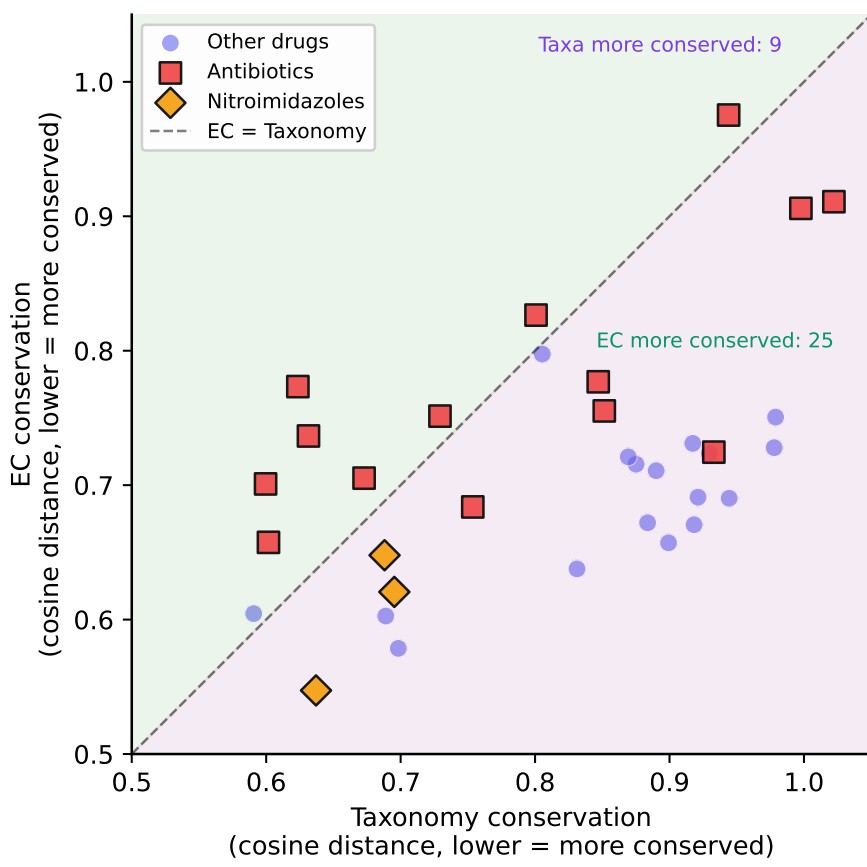

Figure 1: Per-drug conservation across communities. Points below the diagonal have more conserved EC profiles; points above have more conserved taxonomy profiles. Antibiotics (red squares) cluster in the taxonomy-conserved region, except nitroimidazoles (orange diamonds), which are prodrugs requiring metabolic activation.

the scientific question. Our results highlight cross-context generalization as a practical criterion for evaluating representation meaningfulness.

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

## A  DATA ACQUISITION AND PROCESSING

**Raw data download.** Paired-end 16S rRNA amplicon sequencing data were downloaded from the European Nucleotide Archive (ENA) under accession PRJNA1297205 (Shi et al., 2025). The dataset comprises 4,092 samples across 9 synthetic intestinal communities (SIC0–8), with FASTQs retrieved via the ENA file report API.

**Amplicon processing.** Raw reads were processed using nf-core/ampliseq (v2.6.1). Forward reads were truncated at 240 bp and reverse reads at 180 bp. The pipeline performs quality filtering, denoising via DADA2, chimera removal, and taxonomic assignment against the SILVA 138 database. Output: ASV count tables with taxonomic annotations at genus level.

**Functional prediction.** Enzyme Commission (EC) profiles were predicted from ASV tables using PICRUSt2 v2.6.3. PICRUSt2 infers functional content via phylogenetic placement against reference genomes with known gene content. Coverage: 99.4% of ASVs successfully placed.

**Drug metadata extraction.** Sample metadata including drug treatment conditions were parsed from ENA `library_name` fields. Each library name follows the pattern `ExpNN-YYMMDD-SICx_condition_Well_Drug_Conc`, from which the `Exp_condition` (e.g., "Rifampicin-10") was extracted via regex.

**ATC code mapping.** Drug names were mapped to Anatomical Therapeutic Chemical (ATC) codes via the PubChem REST API. For each drug name: (1) query PubChem for compound CID, (2) retrieve pharmacology annotations including ATC codes. ATC Level 1 = first character (therapeutic area, e.g., J = Anti-infectives), Level 2 = first 3 characters (pharmacological subgroup, e.g., J01 = Antibacterials).

## B  MODEL DETAILS

**Delta profiles.** For each sample: (1) aggregate ASV counts to genus level, (2) compute relative abundances, (3) apply centered log-ratio (CLR) transform: $\mathrm{CLR}(x_i) = \log(x_i) - \frac{1}{D}\sum_j \log(x_j)$, (4) subtract control centroid (mean of untreated samples in that community). This yields "delta profiles" capturing drug-induced perturbations relative to baseline.

**PCA.** Standard PCA retaining 10 components, fit on standardized (zero-mean, unit-variance) delta profiles.

**VAE.** $\beta$-VAE with encoder 256→128→64→10 (latent), symmetric decoder, $\beta$=0.5 (KL weight), trained with Adam (lr=$10^{-3}$) for 200 epochs. Architecture selected from 192-configuration sweep on held-out SIC0 drugs.

**MGM.** Microbiome Generative Model (Zhang et al., 2025): a GPT-2 architecture pretrained on 263K microbiome samples. We format our abundance data as taxon-count sequences, tokenize using the MGM tokenizer, and extract the final hidden state (256-d) as the sample embedding.

## C  RETRIEVAL EVALUATION

**Task formulation.** Given drug embeddings, evaluate whether drugs with similar embeddings share the same mechanism of action. This is a retrieval task, not classification.

**MAP@10 computation.** For each query drug $q$:

1. Compute pairwise Euclidean distances to all other drugs
2. Retrieve the $k$=10 nearest neighbors
3. For each neighbor, check if it shares $q$'s ATC class (binary relevance)
4. Compute Average Precision: AP $= \frac{1}{R} \sum_{i=1}^{k} P(i) \cdot \text{rel}(i)$, where $R$ = number of relevant items, $P(i)$ = precision at rank $i$, $\text{rel}(i) = 1$ if item $i$ is relevant

Mean Average Precision (MAP@10) = mean AP over all query drugs.

**Within-community evaluation.** On SIC0: 70/30 drug-level split (stratified by ATC Level 2). Train VAE/PCA on training drugs, evaluate on held-out drugs. 101 drugs in test set.

**LOOCV evaluation.** Leave-one-community-out on SIC1–8: for each fold $i$, hold out SIC$_i$ as test, train on remaining 7 communities. VAE and PCA retrained each fold; hyperparameters fixed from SIC0 tuning. Drug-level embeddings computed by averaging sample-level embeddings within each drug. Reports mean $\pm$ std over 8 folds.

**Statistical testing.** Paired bootstrap ($n$=10,000) over the 8 LOOCV folds to test whether one representation significantly outperforms another.

