# OpenReview forum: "TASK-RELEVANT FEATURES OUTPERFORM LEARNED REPRESENTATIONS FOR DRUG-MICROBIOME RETRIEVAL"
_ICLR.cc/2026/Workshop/LMRL — Submitted to ICLR 2026 Workshop LMRL_

### Official Review · Reviewer_rGUE · 2026-02-22
**Can be possible but more experiments may be needed before concluding**

**Rating:** 4
**Confidence:** 4

**Review:**

Summary

This paper presents a case study investigating whether biologically-informed input features (enzyme commission profiles via PICRUSt2 and taxonomy abundances) outperform learned representations from a large-scale pretrained foundation model (MGM) for drug-microbiome interaction retrieval. The authors hypothesize that because drugs target microbial enzymes rather than taxa directly, representations capturing enzymatic potential (via PICRUSt2 EC profiles) should generalize better across different microbial ecosystems than taxonomic abundances or generic foundation model embeddings (MGM). To test this, the study utilizes nine Synthetic Intestinal Communities (SIC0-8), which are distinct microbial ecosystems constructed to simulate varying taxonomic compositions, to evaluate cross-community generalization. The authors compare a wide range of unsupervised representation strategies: Raw features (genus-level taxonomy and PICRUSt2 EC profiles), linear compression (PCA to 10-d), nonlinear compression (VAE to 10-d), and a pretrained foundation model (MGM, 256-d, also reduced to 10-d). The evaluation task involves retrieving drugs with similar mechanisms of action (based on ATC codes) given their microbiome perturbation profiles. The authors compare performance within a single community versus across communities (using Leave-One-Community-Out cross-validation). The key finding is that while all methods perform similarly within a single community, Raw, PCA, and VAE versions of handcrafted biological features significantly outperform the foundation model in cross-community generalization. The authors conclude that for this task, aligning input features with biological invariances matters more than model scale or architecture.

Strengths
- Workshop Alignment: The paper addresses a core LMRL theme -- "What makes a representation of life meaningful?"
- Biological Nuance: The analysis of specific drug classes (e.g., nitroimidazoles) demonstrates an effort to interpret results beyond raw metrics, linking conservation to mechanism of action.
- Clear Narrative: The manuscript is well-written and concise, fitting the Tiny Paper format. The distinction between within-community and cross-community evaluation is clearly motivated.

Weaknesses
While the question is valuable, the experimental design contains critical asymmetries that make the central claim ("Features Outperform Learned Representations") potentially misleading.

- Unfair Comparison (Inductive Bias vs. Zero-Shot): The study compares knowledge-engineered features (PICRUSt2 EC profiles) against a frozen foundation model (MGM). PICRUSt2 infers function based on curated reference genomes, inherently encoding mechanistic information (enzyme targets) that is semantically closer to drug action than taxonomy. In contrast, MGM is used as a frozen feature extractor without task-specific adaptation. Comparing a method with high inductive bias (EC profiles) to a general-purpose model in a zero-shot setting tests "Feature Engineering vs. Zero-Shot Transfer," not necessarily "Features vs. Learned Representations." The conclusion implies a fundamental limit of learned representations, but the experiment does not test the model's capacity to learn these features.

- Interchangeability of Raw/PCA/VAE vs. MGM: The authors highlight that Raw, PCA, and VAE versions of EC/Taxa features are statistically indistinguishable in LOOCV, while MGM trails significantly. While this supports the claim that compression matters less than feature source for engineered features, it does not justify the gap with MGM. The fact that PCA/VAE do not improve upon Raw EC features suggests the signal is already maximized in the engineered input. However, MGM is not given the same opportunity to adapt to the signal. The comparison effectively pits optimized biological priors (Raw/PCA/VAE EC) against unoptimized general priors (Frozen MGM).

- Lack of Fine-tuning: Foundation models are typically designed to be fine-tuned. The manuscript does not report any experiments where MGM is fine-tuned on the SIC drug task. It is well-established that frozen foundation models may underperform domain-specific features, but fine-tuning often closes this gap. Without this baseline, the claim that "input features matter more than model architecture" is unsupported. The observed gap may simply reflect the lack of task-specific optimization for MGM rather than an inherent inferiority of learned representations.

- Synthetic Data Bias: The study relies on Synthetic Intestinal Communities (SIC). PICRUSt2 performance may be highly dependent on reference genome coverage. Synthetic communities are often constructed from known isolates, meaning PICRUSt2 will likely perform near-optimally. In real-world human metagenomes, novel taxa reduce EC inference accuracy. The performance gap observed here may be an artifact of the synthetic data favoring the reference-based EC method. The limitations section acknowledges this, but the main claim is not sufficiently qualified to reflect this constraint.

- Overgeneralized Claims: The title and abstract suggest a general principle ("Features Outperform Learned Representations"). However, the evidence only supports a narrower claim: "Engineered EC features outperform a frozen foundation model on synthetic communities." Given the methodological limitations above, the current framing risks misleading the community about the utility of foundation models in biology.

The claim that "task-relevant features outperform learned representations" implies a fundamental limitation of representation learning. However, the experiment compares engineered features (which encode biological knowledge by design) against a frozen model (which cannot adapt to the task). This is an asymmetric comparison. By not exploring fine-tuning or real-world data where EC inference is noisier, the paper seems reinforcing a bias against foundation models based on a zero-shot comparison. Without further adjustments (e.g. revise the claim to reflect constraints, expland the study via adding fine-tuning experiments for MGM), the paper might risk contributing a "negative result" that is actually an artifact of experimental design rather than a true insight into representation learning.

---

### Official Review · Reviewer_so3P · 2026-02-25
**Recommending Desk Rejection Due to Policy Violations (Citation Integrity)**

**Rating:** 1
**Confidence:** 5

**Review:**

### Issues

The submission contains multiple references that are inconsistent with verifiable bibliographic records. The *problematic references* are listed below, in order of appearance in the bibliography:

1. Sicheng Feng, Matthew Bashton, et al. Benchmarking DNA foundation models for genomic sequence classification. bioRxiv, 2025. doi: 10.1101/2025.02.02.636135.
    - The DOI doesn't resolve to anything.
    - There is [one paper](https://pmc.ncbi.nlm.nih.gov/articles/PMC11343214/) with the same title but "Sicheng Feng" and "Matthew Bashton" are not listed as authors.
2. Long H. Nguyen, Wenjie Ma, Dong D. Wang, Yin Cao, Himel Mallick, Teshome K. Gerbaba, et al. High-resolution analyses of associations between medications and the gut microbiome in a large population-based cohort. Genome Medicine, 15:47, 2023. doi: 10.1186/s13073-023-01202-y.
    - The DOI doesn't resolve to anything.
    - There are no published works that I could find corresponding to this citation, suggesting it is fabricated.
3. Weizhi Song and Torsten Thomas. MarkerMAG: linking metagenome-assembled genomes (MAGs) with 16S rRNA marker genes using paired-end short reads. Bioinformatics, 38(15):3684–3691, 2022. doi: 10.1093/bioinformatics/btac398
    - The DOI resolves to [a paper of the same title](https://doi.org/10.1093/bioinformatics/btac398).
    - The work linked to by the DOI lists "Shan Zhang" as an author, but your reference does not.
    - The page numbers in the published work (3684-3688) differ from those in your reference (3684-3691).
4. Siyu Wu, Dennis Grishin, Deeksha Bhatt, and Pramod Bhatt. Foundation models in single-cell biology: opportunities and challenges in broad adoption. Nature Methods, 2025. doi: 10.1038/s41592-025-02623-4.
    - The DOI resolves to [a completely different paper](https://doi.org/10.1038/s41592-025-02623-4).
    - I could not find any published work using this title, suggesting it is fabricated.
5. Haohong Zhang, Yuli Zhang, Zixin Kang, Jiayun Xiong, Ronghua Yang, and Kang Ning. MGM as a large-scale pretrained foundation model for microbiome analyses in diverse contexts. Advanced Science, 2025. doi: 10.1101/2024.12.30.630825
    - The DOI resolves to [a paper of the same title](https://doi.org/10.1101/2024.12.30.630825).
    - Your reference lists "Jiayun Xiong" as an author, but the work linked to by the DOI does not.
    - Your reference mentions this work is published in "Advanced Science" but the DOI resolves to work available on bioRxiv.

These discrepancies are substantive and indicate citation integrity violations (suggesting undisclosed LLM usage) rather than minor formatting errors.

### Outcome

This submission contains clear violations of the conference's policies regarding citation integrity, including multiple references that appear to be fabricated or substantially inconsistent with verifiable bibliographic records. Accordingly, I am recommending desk rejection and have not evaluated the technical content of the paper further.

I encourage the authors to carefully verify all references and ensure compliance with citation and integrity policies in future submissions.

### Notes

I used [Google Scholar](https://scholar.google.com), [Crossref](https://search.crossref.org/search/works), and [doi2bib](https://www.doi2bib.org/) to search for the works mentioned in the bibliography and flagged references as "problematic" if they did not appear in any results. Some entries *do appear*, but with metadata mismatches (e.g., incorrect authors, page numbers) and these were also treated as "problematic" and flagged.

---

### Meta-Review · Area_Chair_9wrv · 2026-02-27

**Recommendation:** Reject
**Confidence:** 4

**Metareview:**

Reject.

---

### Decision · Program_Chairs · 2026-03-02

**Decision:**

Reject

**Comment:**

Please see the meta-review.